# Low-Cost Ka-Band Cloud Radar System for Distributed Measurements within the Atmospheric Boundary Layer



**Roberto Aguirre** [1] , **Felipe Toledo** [2] , **Rafael Rodríguez** [3,4] , **Roberto Rondanelli** [5,6,7] , **Nicolas Reyes** [1,8] and **Marcos Díaz** [1,6,*]

1   Electrical Engineering Department, Faculty of Physical and Mathematical Sciences, University of Chile, Santiago 8370451, Chile; roberto.aguirre@ug.uchile.cl (R.A.); nireyes@u.uchile.cl (N.R.)
2   Laboratoire de Météorologie Dynamique, École Polytechnique, Institut Polytechnique de Paris, 91128 Palaiseau, France; ftoledo@lmd.polytechnique.fr
3   Institute of Electricity and Electronics, Facultad de Ciencias de la Ingeniería, Universidad Austral, General Lagos 2086, Valdivia 5110701, Chile; rafael.rodriguez@uach.cl
4   CePIA, Astronomy Department, Universidad de Concepción, Casilla 160-C, Concepción 4030000, Chile
5   Department of Geophysics, Faculty of Physical and Mathematical Sciences, University of Chile, Santiago 8370449, Chile; ronda@dgf.uchile.cl
6   Space and Planetary Exploration Laboratory, Faculty of Physical and Mathematical Sciences, University of Chile, Santiago 8370451, Chile
7   Center for Climate and Resilience Research, Faculty of Physical and Mathematical Sciences, University of Chile, Santiago 8370449, Chile
8   Max Planck Institute for Radioastronomy, Auf dem Hugel 69, 53121 Bonn, Germany
*   Correspondence: mdiazq@ing.uchile.cl; Tel.: +56-2-2978-4204

**Abstract:** Radars are used to retrieve physical parameters related to clouds and fog. With these measurements, models can be developed for several application fields such as climate, agriculture, aviation, energy, and astronomy. In Chile, coastal fog and low marine stratus intersect the coastal topography, forming a thick fog essential to sustain coastal ecosystems. This phenomenon motivates the development of cloud radars to boost scientific research. In this article, we present the design of a Ka-band cloud radar and the experiments that prove its operation. The radar uses a frequency-modulated continuous-wave with a carrier frequency of 38 GHz. By using a drone and a commercial Lidar, we were able to verify that the radar can measure reflectivities in the order of $-60$ dBZ at 500 m of distance, with a range resolution of 20 m. The lower needed range coverage imposed by our case of study enabled a significant reduction of the instrument cost compared to existent alternatives. The portability and low-cost of the designed instrument enable its implementation in a distributed manner along the coastal mountain range, as well as its use in medium-size aerial vehicles or balloons to study higher layers. The main features, limitations, and possible improvements to the current instrument are discussed.

**Keywords:** cloud radar; boundary layer; frequency modulated continuous wave radar; Ka-band radar

## 1. Introduction

Chile has topographic characteristics that allow the study of coastal fog and its relation with the boundary layer. Along the Chilean coastal range fog intersects with the mountain topography, Figure 1 shows an example captured about 20 km west from Santiago, the capital of Chile, in *Lo Prado* tunnel, where it is possible to see a cloud coming from the west intersecting with the central Chile coastal

range. Close interaction with coastal fog facilitates studies not only of macro characteristics of clouds, but also allows the study of its microphysics by using in situ measurements with multiple instruments.

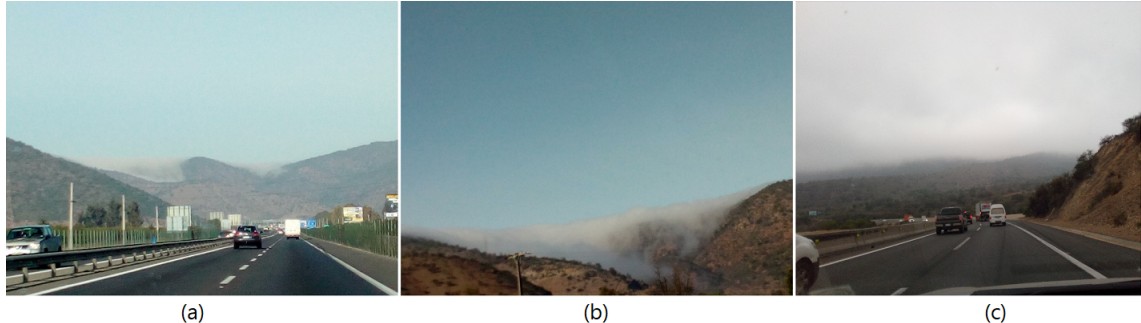

|(a)|(b)|(c)|

**Figure 1.** Example of a cloud intersecting with the Chilean coastal range in *Lo Prado* tunnel, west from Santiago, Chile's capital. (**a**) shows the east side of the tunnel. It is possible to see a clear sky at that side of the range; (**b**) shows a close up to the top of the mountain over the tunnel, where the cloud is appearing; (**c**) presents the situation at the west side of the tunnel, where there is a cloudy sky. Images were taken on 11 November 2019, in trip from Santiago to Valparaiso, before and after crossing the tunnel.

The possibility of interception of clouds with the coastal range motivates the establishment of an in situ regional fog observatory to obtain continuous measurements of cloud properties in the Fray Jorge national park, north of Chile. Continuous measurements can improve the understanding of physical interactions driving the fog life cycle, improving the quality of fog forecast models [1]. These models have a wide range of applications such as climate [2], aeronautics [3], and solar energy studies due to the effect of clouds in Earth radiation budget [4] and in astronomy not only for site testing, but also for observatories operation [5]. In addition, this facility is expected to serve as a calibration site for new developed remote sensing instruments that might be later used along the coastal range, as well as to serve as either calibration or ground truth for satellite products that estimate physical parameters of clouds in larger areas.

Remote sensing instruments such as Radars and Lidars are key to study large scale phenomena such as clouds that defy conventional surface and upper air meteorological observations [6]. Cloud radars can estimate profiles of cloud and fog properties such as reflectivity, scatterers doppler speed, liquid water and ice content, and precipitation rate [7–11]. The combination of both ceilometer (Lidar) and cloud radar can provide relevant information of the internal structure of clouds as well as the boundaries of a cloud layer during continuous operational use. The ceilometer is very precise at detecting the base of a cloud layer, but cannot commonly detect the cloud top due to attenuation of the beam in the cloud. Meanwhile, cloud radars are able to penetrate the cloud, giving internal information as well as detecting the cloud top depending on the range limit of the radar and the thickness of the cloud. In addition, radars could eventually detect higher layers in the case of multilayered clouds. Cloud lower and upper limits are important to study the impact of clouds in a changing climate. High-resolution observations of the cloud and fog boundaries can help modelers verify and improve local fog prediction models or numerical weather prediction climate models. There is also a general need for increasing automatic cloud observations for climate studies at monitoring laboratories located in the coastal range of Chile, where the understanding of fog and low clouds behavior remains a relevant challenge.

Although the performance of current commercial radars is adequate, the size, power consumption, and cost of them inhibit its use to study the fog dynamics in the mesoscale or as network instruments. The use of frequency-modulated continuous-wave (FMCW) radar has significantly reduced the cost of the system with a low impact on performance [12]. However, current commercial FMCW radars are still large in size, require hundreds of watts in power, and cost hundreds of thousands of dollars, which is still a restrictive solution to cover the large areas of the Chilean coastal range. For this

reason, taking advantage of the Chilean topography, we proposed a miniaturized FMCW radar much cheaper than pulsed radars and lower in cost compared to current cloud commercial FMCW radars [13]. Although for this work we used recycled radio astronomy components, these components are nowadays available commercially for satellite communication. The current cost of our proposed radar is less than 20,000 US dollars. The reduction in size and power consumption is also key to facilitate the location and maintenance of the instruments at isolated locations in the coastal range. These features offer the possibility of using a larger number of instruments, allowing multi-point measurements. In our first attempt of the compact radar, the range resolution was larger than the range allowing just a pixel in the measurement [13]. In this article, we present the changes to the first radar prototype to improve the height estimation of the cloud base, which allows for a better characterization of clouds within the boundary layer. In addition, we present a novel procedure for distance calibration by using a drone. By using this procedure together with an internal calibration of the radar, we conclude that the new version of our system possesses a range coverage over 500 m with a range resolution of ∼20 m. Finally, we present the results of a field campaign, where simultaneous measurements of a cloud are taken with the radar in and a commercial ceilometer. In addition, we present MODIS data of the area together with radiosonde data. The sounding was launched from Santo Domingo, about 120 km south from our field campaign, about two hours before of our Radar and Lidar measurements. This type of radar not only can operate alone but might complement commercial ceilometer. In [14], it was shown that FMCW radar measuring simultaneously with a ceilometer provides multiple advantages. The combination of both ceilometer and FMCW cloud radar can provide the boundaries of a cloud layer in a cost effective manner during continuous operational use in a coastal cloud laboratory taking advantage of the proximity of clouds.

## 2. Cloud Radar Prototype

### 2.1. Measurement Principle

The radar developed in this work is of the Frequency-Modulated Continuous-Wave (FMCW) type, similar in principle to the cloud radar described in [12]. As in most linearly modulated radars, an up-chirp waveform is transmitted, while the receiver collects the echo signal (see Figure 2). It follows that the range (r) to a target is proportional to the frequency difference between transmitted and received signals following Equation (1):

$$r = \frac{F_d T_{rep} c}{2B},$$ (1)

where $F_d$ is the frequency difference, $T_{rep}$ is called the repetition period, c is the speed of light, and $B$ is the modulation bandwidth.

For FMCW radars, the range resolution ($\Delta r$), the minimum discernible distance between targets, depends on the modulation bandwidth according to Equation (2). However, the actual range resolution can be degraded due to non-ideal factors in FMCW radars [15,16]:

$$\Delta r = \frac{c}{2B}.$$ (2)

The wider the bandwidth, finer cloud structures could be resolved in the radar height profiles. On the other hand, the maximum unambiguous range ($r_a = T_{rep} \cdot c/2$) depends on the repetition period. Therefore, the modulation bandwidth and the transmitted power have to be selected properly to guarantee enough points inside the unambiguous range.

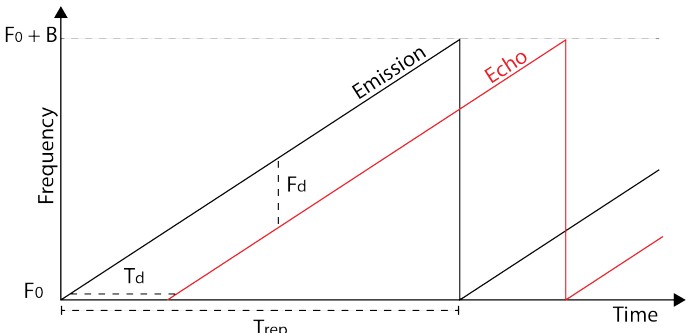

**Figure 2.** Transmitted up-chirp radar waveform and the corresponding echo signal arriving at the receiver from a single stationary point target. If multiple targets are present at different ranges, $F_d$ is the linear combination of the response for each point target.

For cloud radars operating within the Rayleigh Scattering Regime, the received power $P_r$ from a target at range $r$ is related to the target radar reflectivity ($Z$) by the meteorological radar equation as presented in [17]:

$$10 \cdot log(P_r) = 10 \cdot log(Z) - 20 \cdot log(r) + C_{radar}, \tag{3}$$

where the radar calibration constant ($C_{radar}$) depends on the radar parameters and the dielectric characteristics of the target. The calibration constant specific to this radar will be estimated in Section 3.2.

### 2.2. Radar Hardware

The block diagram of the radar system is presented in Figure 3. The *transmitter* continuously feeds the transmitter antenna with an up-chirp waveform within the Ka-band, produced by modulating a Gunn oscillator with a saw-tooth reference signal. The receiver antenna delivers the echo-signal to the *receiver*, which down-converts the input signal by multiplying it with the currently transmitting signal, and directs the resulting intermediate frequency (IF) signal to a digital acquisition system (DAS). The DAS converts the IF signal and transmitter reference voltage (see scheme in Figure 3) into digital signals to compute reflectivity and Doppler profiles by using a MATLAB script in a desktop computer.

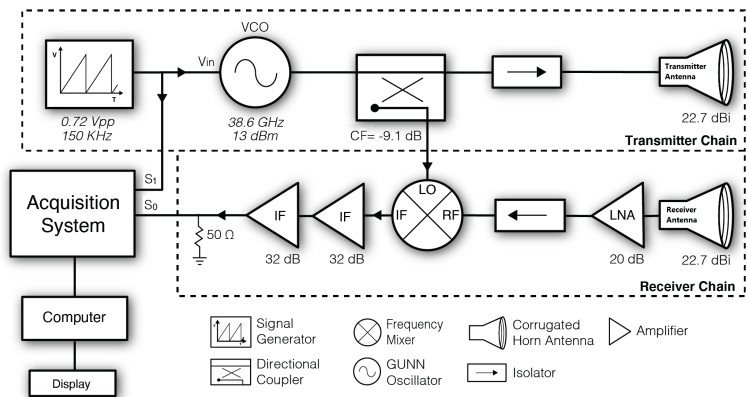

**Figure 3.** FMCW Radar Block diagram.

The ideal transmitted chirp waveform is repeated with a frequency of 150 kHz, yielding a theoretical non-ambiguous range of 500 m (see Equation (1), with $F_d = B$). The chirp bandwidth is set to 10 MHz, thus producing, at best, a range resolution of 15 m (see Equation (2)).

The DAS analog-to-digital-converters (ADC) acquire $10^7$ samples, every 10 s for cloud measurements or every 3 s during the drone experiment, with a sample-rate of 20 MHz, and with a range of $\pm 1$ V.

Implementation specific details of the transmitter chain, antennas, receiver chain, and Data Acquisition System can be found in Appendix A.

Figure 4 shows the actual implementation of the transmitter chain, receiver chain, and power supply of the radar system.

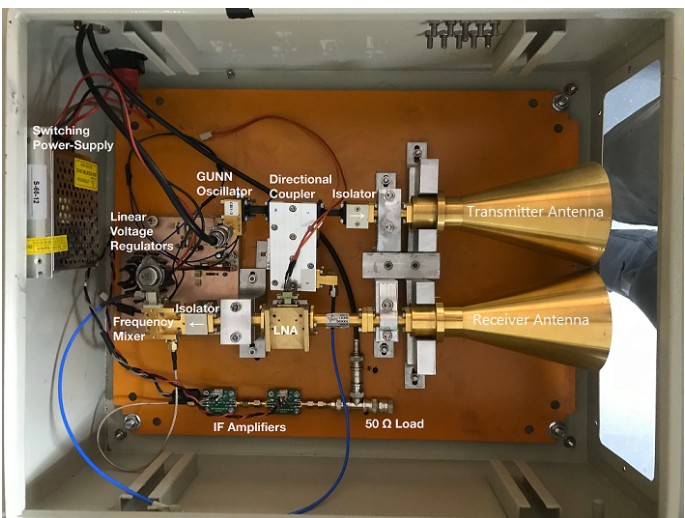

**Figure 4.** The actual radar. In the image, it is possible to see the power supply and the Transmitter and Receiver chains. The container box is a conductor which is usually employed in the implementation of electrical panels. It serves as a structure to attach the components and also provides protection from the environment during field campaigns. During operation, the aperture in front of the antennas is covered with a material that is practically transparent within the Ka-band.

*2.3. Digital Processing*

This section describes how we obtain a reflectivity profile using the signals acquired by our radar. This is an implementation of the pulse-pair processing algorithm. This procedure reduces the background noise implementing a coherent integration trough time, and it also allows the retrieval of Doppler profiles by comparing the phase of sequential pulses. The same signal processing step is being used in other FMCW cloud radars such as [12,18]. For this algorithm to work, we need to have correlated pulses. The biggest challenge to this requirement is the lack of a phased-locked-loop for the local oscillator. Analyzing the oscillator's stability, we observed a variation of 8 KHz during 80 μs, and this was considered acceptably low given the repetition interval of 6.6 μs.

Because we have a homodyne receiver, the back-end signal frequency ($S_0$ in Figure 3) equals the frequency difference between transmitted and received signals. This information allows for the retrieval of target's range with Equation (1). The transmitter reference voltage ($S_1$ in Figure 3) will be used to find the start of each repetition period. By only processing the second half of each repetition period, we reduce ambiguous range measurements, in exchange of bandwidth.

The transmitter reference signal has the shape of a saw-tooth. The discrete derivative determines the slope, thus the saw-tooth rising edges (pulse start) are found by searching for transitions between a negative and positive slope. For each half-period, the Fast Fourier transform (FFT) of the IF signal ($S_0$ in Figure 3) weighted by a Hann window is calculated to find the impulse response ($R$) of the IF signal. Thus,

$$R = FFT\left(IF \cdot Hann\right). \tag{4}$$

The spectral density function is obtained by multiplying in pairs the sequence of $N$ radar pulses in frequency ($R$) following:

$$X = \frac{1}{N} \sum_{i=1}^{N-1} conj(R_i) \cdot R_{i+1}. \tag{5}$$

Then, the module of $X$ represents the echo received power and the complex argument of $X$ is proportional to the Doppler velocity, as in [12]. This uncorrected Doppler velocity will be presented alongside power or reflectivity profiles for qualitative analysis.

A detection threshold is taken as the average power coming from the penultimate range bin plus 2 dB. At that range, we predict just noise because the echo power is expected to be below the minimum-detectable-signal. For the range cells where the echo power is higher than this threshold, we find the radar reflectivity ($Z$) using the meteorological radar Equation (3) [17], where $Z$ in dB is obtained by using $|X|$ as the $10 \cdot log(P_r)$, after subtracting the noise mask power.

## 3. Calibration Procedures

In this section, we present a calibration procedure, which was designed and implemented to improve the range estimation of the radar. In addition, we estimate the radar calibration constant ($R_C$) for the reflectivity equation (Equation (3)), which allows a more precise estimation of the reflectivity ($Z$). The effectiveness of the procedures will be evaluated by a field campaign with the radar.

### 3.1. Distance Calibration with an Unmanned Aerial Vehicle

The distance resolution $\Delta r$ of FMCW radars is calculated using the chirp bandwidth $B$. An error in the estimation of this term introduces a bias in the range estimation proportional to the target distance. In addition, uncorrected time delays in the microwave components can introduce a constant offset in the beat frequency compared to what is expected from the physical time of flight, and this can produce a constant bias in distance estimation.

To avoid biases in the distance estimation, we implement an external range calibration experiment based on the method published by Feng et al. [19]. This method is developed for FMCW radars and consists in using external targets placed at known distances from the radar to calibrate the relationship between the IF frequency $F_d$ and range retrievals.

The target used in this experiment is a commercial Unmanned Aerial Vehicle (UAV) Xiaomi MI. The UAV is programmed to hover at three known heights above the radar (200, 300, and 400 m), using its internal GPS to control its position. Each position is held for 45 s, to guarantee an accurate estimation of the $F_d$ frequency associated with each radar position. The experimental setup is presented in Figure 5.

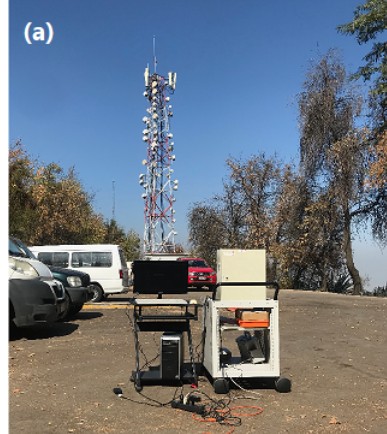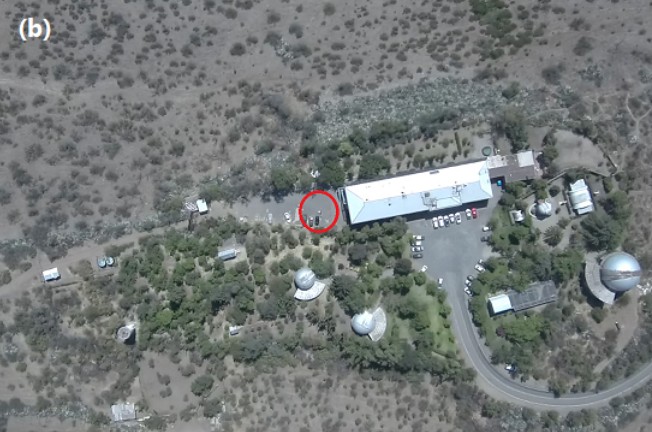

**Figure 5.** Image of the distance calibration procedure, which uses a UAV as target and the internal GPS of the UAV to estimate its height. (**a**) The radar setup in the Astronomy Campus at the University of Chile (33°23′48″S, 70°32′13.3″W). (**b**) Picture taken by the UAV during the procedure.

Figure 6a shows the raw received power versus distance for the complete experiment and their associated Uncalibrated Doppler Velocities. Valid measurements must have a power value 2 dB higher than the noise floor, sampled using the power at the penultimate range bin where no external signal is

observed. Uncalibrated Doppler velocity is proportional to physical velocity, and it is positive when the target moves towards the radar.

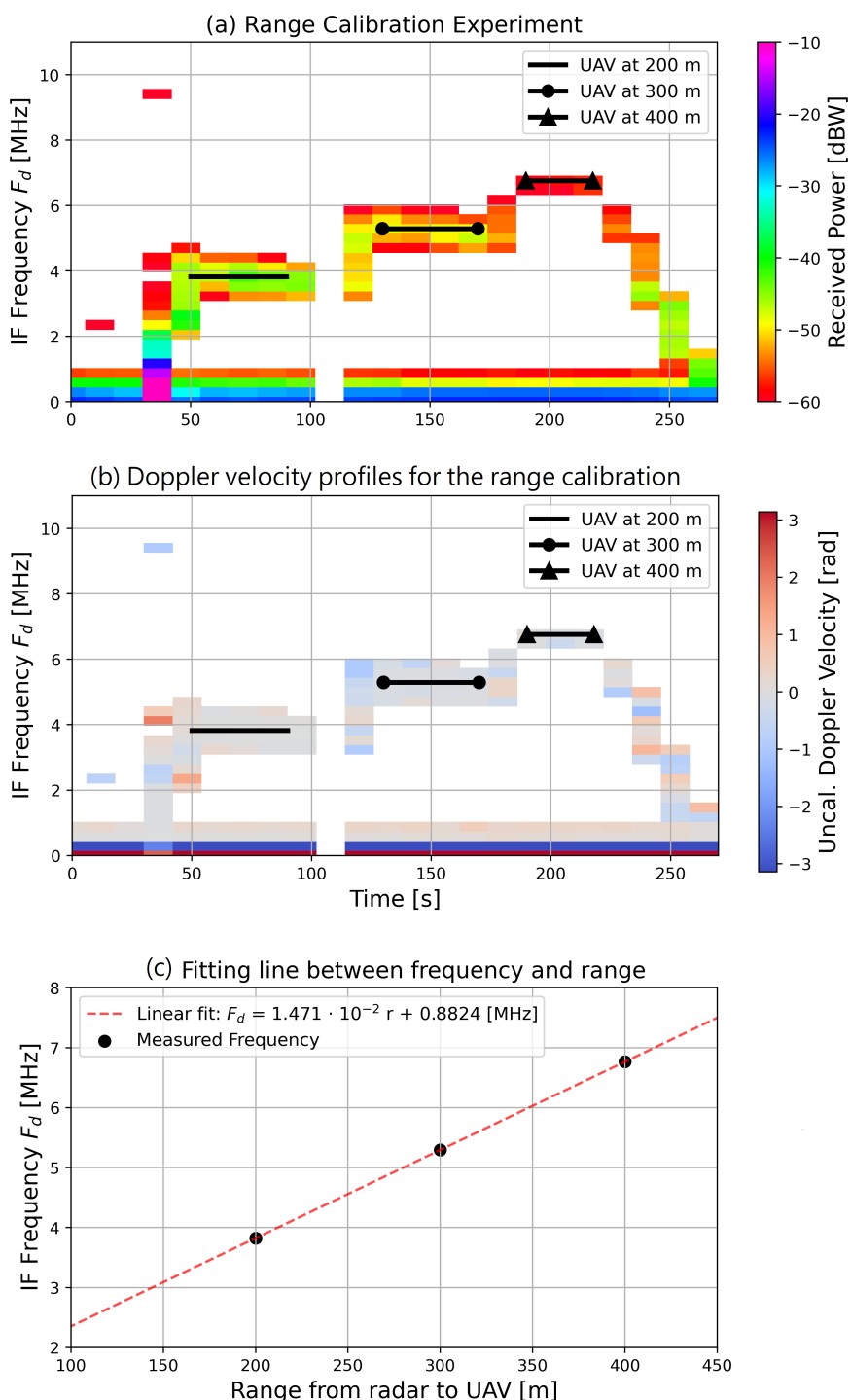

**Figure 6.** Distance calibration using a UAV as the range reference. (**a**) Vertical Raw Power profile for the range calibration experiment. (**b**) Doppler velocity profile. Positive velocities indicate movement towards the radar. Time is in seconds from the begining of the Radar measurements. The UAV hovering height versus time is indicated with the black lines; (**c**) Fitting line between the IF frequency $F_d$ and the UAV position.

Table 1 summarizes the results from the calibration experiment. To estimate the $F_d$ value associated with each UAV position, we select the profiles within the hovering periods. Then, we retrieve for each profile a single IF frequency associated with the profile maximum power. Finally, we take the median to estimate $F_d$ for the UAV range. Results are shown in Table 1.

**Table 1.** Results of the distance calibration experiment.

| Distance from Radar to UAV [m] | Backscatter Signal IF [MHz] |
| --- | --- |
| 200 | 3.8235 |
| 300 | 5.2941 |
| 400 | 6.7647 |

The results from Table 1 are related using a linear fit, as shown in Figure 6b. There we observe that the radar follows a linear relationship between frequency and range, as expected from theory (see Equation (1)). However, the slope of $1.471 \cdot 10^{-2}$ [MHz m$^{-1}$] retrieved experimentally is larger than what is expected from theory ($\approx 1.00^{-2}$ [MHz m$^{-1}$]). Additionally, we also observe an offset. With this data, we calculate the coefficients needed to get the range $r$ for each $F_d$ value [19], obtaining the relationship of Equation (6):

$$r[m] = \frac{F_d[MHz] - 0.8824}{1.471 \cdot 10^{-2}}. \tag{6}$$

With this calibration relationship and considering the $F_d$ resolution (300 kHz), we obtain a range resolution ($\Delta$r) of 20 m.

### 3.2. Internal Calibration

The calibration constant ($C_{radar}$) is required for retrieving reflectivity (see Equation (3)). $C_{radar}$ can be measured precisely with an already characterized reflector. Since none were available, we performed an internal calibration. This calibration is performed by characterizing each radar element and cascading their effect into the parameters of Equation (7), which is an adaptation of [20] for FMCW radars. Each device signature is found using either a vector network analyzer, a spectrum analyzer, or a finite element method solver as for the radar antennas. The principal parameters are summarized in Table 2. Using those values, the calibration constant for reflectivity is $-109$ dB [$\frac{mm^6}{mW\,m^5}$]:

$$C_{radar} = 10 \cdot log_{10} \left( \frac{512 \cdot ln(2) \lambda^2 10^{18} L_{sys}}{P_t G_a^2 \Delta r \pi^3 \phi^2 \, |K|^2} \right). \tag{7}$$

**Table 2.** Radar parameters for internal calibration. The system losses are negative because they include the receiver gain.

| Parameter | Variable | Value |
| --- | --- | --- |
| Wavelength | $\lambda$ | 7.77 mm |
| Transmitted power | $P_t$ | 12 dBm |
| System losses | $L_{sys}$ | $-67.5$ dB |
| Antenna gain | $G_a$ | 22.7 dBi |
| Beamwidth (3 dB) | $\phi$ | 15 degrees |
| Range resolution | $\Delta r$ | 20 m |
| Index of refraction of water sphere | $|K|^2$ | 0.93 [17] |

## 4. Field Campaign

To evaluate the calibration procedures, which define the performance of the radar, a field campaign for the radar was implemented by comparing its response with the response obtained with a pattern instrument, a Lidar (VAISALA CL31 Ceilometer, e.g., [21]), which provides range corrected attenuated backscatter profiles at 910 nm, with a resolution of 5 m and up to 7.6 km. The campaign took place in the coast of Papudo, Chile (Location of the experiment in GPS coordinates: 32°30′09.7″S, 71°27′35.6″W), on 20 January 2019. Figure 7 (Left) and (Right) shows the experiment setup and an aerial view of the experiment taken by the drone, respectively.

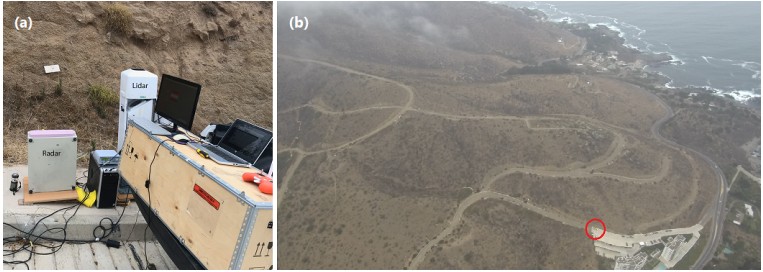

**Figure 7.** (**a**) the experimental setup for measuring clouds with the radar and Lidar; (**b**) image taken with a drone that shows the location of the experiment and in the top left corner you can see marine stratus clouds moving towards the sea. Close to the bottom right corner, the Radar and Lidar position (32°30′09.7″S, 71°27′35.6″W) is marked with a red circle.

Figure 8 shows the Lidar height profile of the back-scattered intensity (The Lidar back-scattered intensity is extracted from CL-VIEW, which is the Lidar vendor software.) and height profile of the radar reflectivity, from 11:07 a.m., until 11:41 a.m., local time. The Lidar and the radar, after the UAV based distance calibration, detect the cloud base at the same altitude of approximately 400 m (a.g.l).

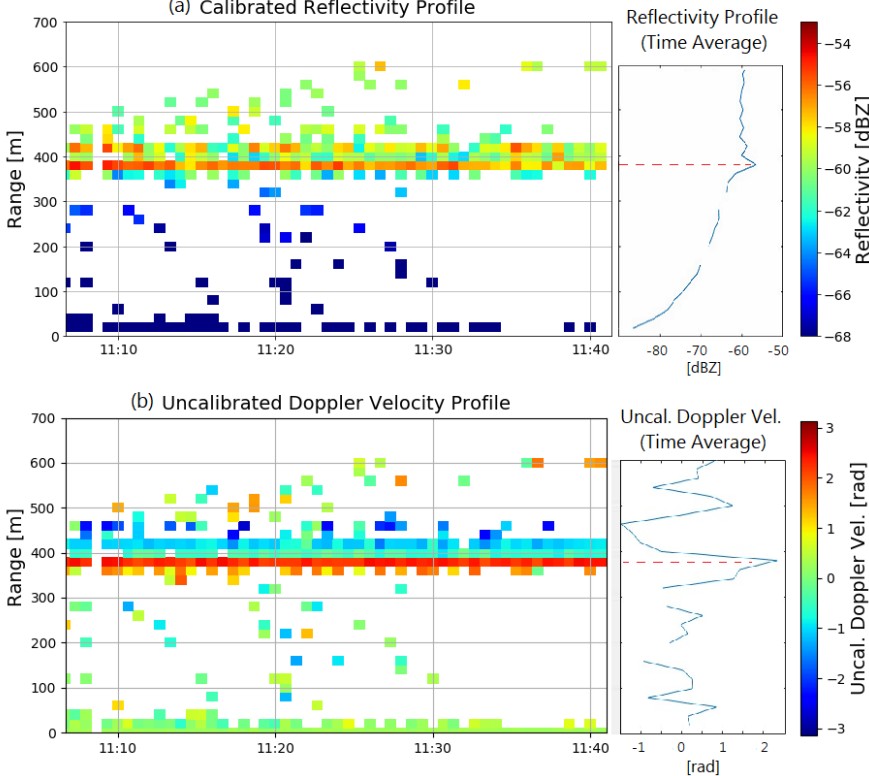

**Figure 8.** *Cont.*

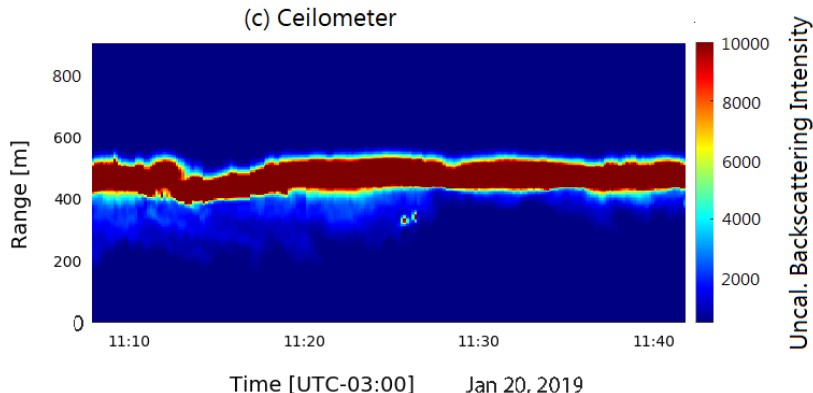

**Figure 8.** (**a**) radar reflectivity retrieval, calculated using the internal calibration and the UAV based distance calibration presented in Section 3. It also shows the time averaged reflectivity profile; (**b**) shows the Uncalibrated Doppler Velocity retrievals. Positive values indicate movement towards the radar. It also shows the time averaged uncalibrated Doppler velocity profile; (**c**) shows the lidar backscattered intensity for the time period from 11:07 a.m. to 11:42 a.m., Chilean local time.

## 5. Discussion

The design of the UAV calibration procedure and the field campaign were designed to be able to evaluate the estimated range limit and range resolution. The field campaign was designed to evaluate the feasibility of using the radar as a portable instrument. The UAV calibration procedure and field campaign showed that this radar design was able to retain the advantages envisioned in the original design presented by [13], but overcoming the problems found in that attempt. Now, the unambiguous range is increased to 500 m and the resolution improved to 20 m, which gives over 45 points along the profile.

Although the found resolution of the radar ($\sim$20 m), which was estimated by the calibration procedure, is higher than the theoretically estimated resolution of 15 m, it is known that certain non-ideal factors can degrade the range resolution of FMCW radars. Non-ideal factors may include the reduced effective modulation bandwidth due to time-of-flight delays, the increased main-lobe null-to-null bandwidth when FFT windows are applied, a reduced receiver frequency resolution due to a reduction in the pulse processing time, and nonlinearity of frequency sweeps [15,16]. This issue highlights the need of the external calibration method for range.

During the day of the campaign, a large stratocumulus cloud deck covered the Chilean coast intercepting the topography as seen in the MODIS images (Data available in https://worldview.earthdata. nasa.gov/?v=-97.28884731789906,-50.34962354523486,-48.564583224612235,-25.759096510654174&t= 2019-01-20-T15%3A19%3A46Z&l=Reference_Labels(hidden),Reference_Features(hidden),Coastlines, MODIS_Terra_Cloud_Top_Height_Day(palette=rainbow_2),VIIRS_NOAA20_CorrectedReflectance_ TrueColor(hidden),VIIRS_SNPP_CorrectedReflectance_TrueColor(hidden),MODIS_Aqua_ CorrectedReflectance_TrueColor(hidden),MODIS_Terra_CorrectedReflectance_TrueColor) (Figure 9). From the nearby Santo Domingo upper air sounding station (approx. 120 km south of Papudo also at the coast), at 12:00 UTC (i.e., about 2 h before the measurements presented) the cloud layer was identified from about 310 to 663 m a.m.s.l. (Data available in https://climatologia.meteochile.gob.cl/application/index/productos/RE5014 by using the Santo Domingo station code of 330030 with the time of 12:00 p.m.) (see Figure 10). One could reasonably expect a relatively homogeneous cloud top height near the coast even for the relatively large extension between Santo Domingo and Papudo as seen in the MODIS cloud top temperature image (Figure 9). Cloud top temperatures are within 283 to 285 K (10 to 12 C) over a large region off the coast of Chile, including Santo Domingo and Papudo. The sounding from Santo Domingo confirms that this cloud top temperature corresponds to the temperature at the base of the temperature inversion layer that

extends from 600 m to about 1600 m, therefore confirming that the top of the cloud was at about 600 to 700 m at both sites.

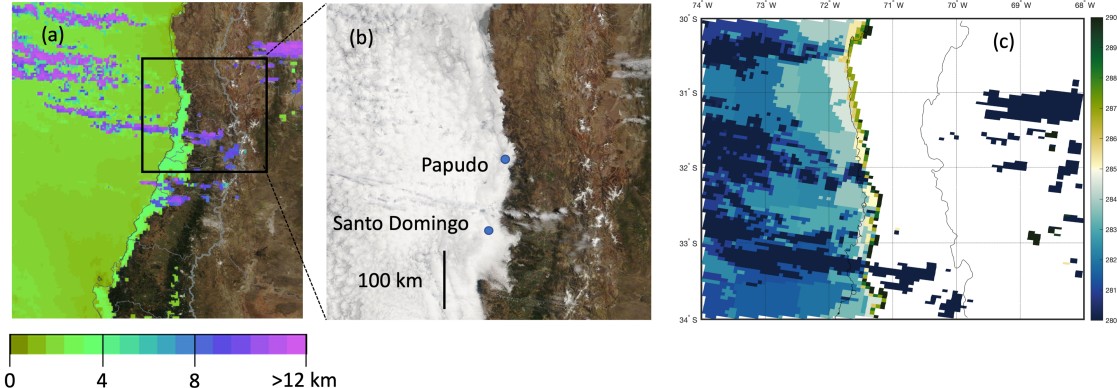

**Figure 9.** (**a**) MODIS terra cloud height on 20 Jan 2019 3:00 p.m. UTC, (**b**) MODIS True Color (Terra, Day) 20 January 2019 3:00 p.m. UTC, and (**c**) MODIS Cloud Top Temperature [K] (Terra, Day) 20 January 2019 3:00 p.m. UTC.

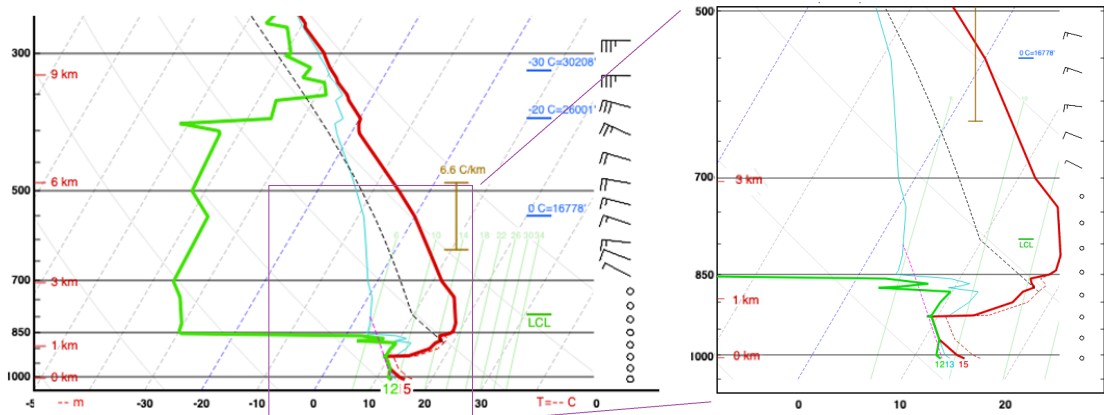

**Figure 10.** Skew-T log-P (SHARPpy) [22]. Sounding launched from Santo Domingo (120 km south of Papudo) on 20 January 2019 at 12:00 p.m. UTC. The red line is temperature in °C and green line is dew point temperature in °C. Cloud is expected when the two curves intercept. Thus, the cloud layer is estimated from 311 to 663 m (a.m.s.l) capped by a strong inversion layer. Data from Dirección Meteorológica de Chile, https://climatologia.meteochile.gob.cl/application/index/productos/RE5014 (Santo Domingo station code of 330030 at 12:00).

Changes during the day also tend to increase the base height even 100 m during the morning hours (e.g., [23]) which makes for a thinner cloud. Therefore, independent observations of the cloud from satellite and radiosonde tend to confirm the cloud base height seen by the cloud radar and ceilometer (∼360–380 m (a.g.l) vs. ∼385–405 m (a.g.l)). Regarding the top of the cloud, this feature should be observed at higher levels than presently seen by the cloud radar which shows very few echoes above 500 m. It is therefore more likely that radar range prevents the characterization of the full structure of the cloud in this case, although some echoes near 600 m could indicate a possible detection of cloud echoes near the top of the cloud.

The lower part of the cloud appears as having a positive Doppler velocity (see Figure 8 (middle-image)), which implies that part of the cloud is approaching the radar. It can be interpreted as drizzle. As expected, the height where Doppler shows a positive velocity is the height with the strongest reflectivity. The internal structure of the stratocumulus cloud as this one is such that the cloud liquid water content should increase linearly between the base and the top of the cloud (e.g., [24]). However, this behavior is not clear in the profile presented in Figure 8a, due to the variations in the higher gates,

which are not only weak echoes but too few. Nevertheless, reflectivity appears to increase from lower to higher gates within the cloud at a rate of 1 to 2 dBZ/100 m, which is consistent with values observed from Stratocumulus over the Southern Pacific (e.g., [25]). However, reflectivity profiles in the presence of light and heavy drizzle are also presented in [25]. In presence of drizzle, the reflectivity reaches its maximum below the cloud base to either decrease or remain constant inside the stratocumulus (Figure 4 in [25]). The reflectivity profile of the field campaign agrees with this description suggesting the presence of drizzle, which is consistent with the Doppler data of the field campaign (Figure 8b). The drizzle might explain the height difference between the radar and the ceilometer. The higher attenuation of drizzle might be responsible for a shorter maximum-detectable-range than expected in the radar (460 m–480 m instead of 500 m).

Figure 6b shows that, if left uncalibrated, the radar would have an error in range retrievals due to a slope different from the theoretical value and an offset. Therefore, the UAV based calibration procedure is essential to guarantee the precision of the range estimation and should be repeated frequently [19].

When comparing Lidar and radar measurements of the cloud base height of the marine stratus shown in Figure 8, after the distance calibration, the obtained difference is less than 40 m. The radar shows reflectivity at lower heights than the ceilometer (consistent with other measurements [25]).

Regarding the stratus observed reflectivity, it is around −55 to −60 dBZ, which is a possible value for clouds at this altitude [26]. These relatively weak reflectivity values indicate that the cloud has a low amount of liquid water content [7]. Low amounts of Liquid Water Content are related to lower optical thickness [27], which is consistent with the good penetration of the Lidar beam in the cloud layer.

Since both instruments observe the same cloud and obtain similar results in distance and cloud thickness, we can conclude that the radar is operative and capable of detecting clouds at least until 500 m of altitude. Still, the sampled cloud has a relatively low reflectivity value and therefore it is likely that the radar can sense thicker clouds at higher altitudes. In any case, the sensitivity of −60 dBZ at 400 m is already enough to study close to the surface phenomena such as fog or low stratus [9,28].

It was discovered that the contamination present in the first four range cells is introduced by the switching power supply ripple through the IF amplifiers. It was noted that, by feeding the amplifiers with a better source, while the rest of the components are operated under the same conditions, the first range cells did not show such noise. Contamination in all ranges, randomly present in both experiments, is attributed to artifacts of the signal acquisition system mentioned in Appendix A.4. These artifacts, when they occur, hinder the detection of the beginning and end of each pulse repetition period, interfering with the spectral density estimation.

Through the development and evaluation of the radar, potential improvements for the radar have been discovered. The difference between the theoretically calculated and experimental range and range resolution estimations are affected by the lack of a Phase-Lock Loop (PLL) for the Gunn oscillator, which can introduce nonlinearities in the frequency sweep. Even if this can be compensated with linearity corrections and frequent UAV range calibration experiments during atmospheric events of interest, the PLL should be implemented in the near future. In addition, having a highly coherent oscillator is also a requirement for an accurate Doppler velocity estimate. It is also recommended to add an internal temperature control system or to build a specific noise model to correct estimations by software. To enlarge the range coverage, we propose to increase the effective area of the antennas. Finally, more capabilities can be added to the radar or better estimation procedures could be implemented in the digital back-end if more customized digital platforms, such as FPGA, are used.

With the current design and campaign, the potentialities of this type of radar were shown. It is able to estimate the base of the cloud with an error of 20 m compared to the estimation of a Lidar. Although this is something that Lidar can do, our system is smaller, consumes less power, and has a lower cost (a ∼2 factor) than the Lidar, although the Lidar can provide more information than just the

base of the cloud. However, the campaign also shows the potentialities of our radar to provide internal information of low altitude clouds or fog. Nevertheless, some improvements have to be performed.

Our radar has shown qualitatively the capability of cloud detection. However, the precision of the radar measurement, especially that inside the cloud, requires a more precise calibration. For instance, a 1 dB error in the calibration can produce 15% error in the liquid water content (LWC) [7]. The necessity of performing standardized calibration is highlighted by [29], by founding up to 2 dB of difference among European radars when comparing with the satellite cloud radar CloudSat. There are more recent works also proposing new calibration analysis and methodologies, embossing the importance of the calibration in the precision of cloud physical parameters estimations that the radar can provide. The calibration can also improve the noise floor estimation. Moreover, Doppler velocity calibration has to be included in the calibration procedure to have precise velocity values. The noise treatment is similar to that used in [12], which requires supervision of the system to be sure that in the penultimate gate range there is not a signal present. To use the radar as an unattended instrument in a coastal range fog laboratory, the noise floor needs to be defined independently of the operation. Longer campaigns closer to the clouds will be the next step to explore the actual performance of the radar in detecting the internal structure of the cloud and ideally the estimation of top limit of it. However, cross-comparison with other radars has to be done outside Chile, since there is no cloud radar in the country.

## 6. Conclusions

The development of national and global cloud radar networks is inhibited by the high cost of these instruments. This work presents a Ka-band FMCW cloud radar design, with an estimated cost under $20,000 USD. Although the current design was developed with recycled components from a radio telescope, many of the components have a commercial substitute that gives the cost estimation. The radar is capable of detecting clouds and estimating their range, up to 500 m of altitude. This range enables the retrieval of fog properties from the base to the top, at least in certain types of topographies such as those found in the coastal range of Chile, at a much lower cost when compared to available cloud radars.

We also present the methods used to calibrate and evaluate the radar. The cloud sampled in the validation campaign with the Lidar and Radar indicates that the Radar prototype operates correctly and is able to detect clouds of approximately $-60$ dBZ of reflectivity at 400 m of distance. This experiment also indicates that the UAV distance calibration was a key step to have correct range measurements with this radar architecture.

The radar is extremely portable and low power compared to commercial clouds radars, facilitating its operation in remote sites. This design is meant to serve as a complement to the ceilometer measurement as well as to work as a distributed radar network to study the dynamics of the boundary layer together with the formation and dissolution of clouds and fog at a mesoscale.

**Author Contributions:** Conceptualization, R.R. (Roberto Rondanelli) and M.D.; Supervision, M.D. and R.R. (Roberto Rondanelli); Investigation, R.A., F.T., and R.R. (Roberto Rondanelli) and M.D.; methodology, R.A., F.T. and R.R. (Rafael Rodriguez); software, R.A. and F.T.; validation R.A. and N.R.; formal analysis, R.A., F.T., M.D., and R.R. (Roberto Rondanelli); resources, N.R., R.R. (Roberto Rondanelli), and M.D.; data curation, R.A., F.T., and M.D.; writing—original draft preparation, R.A. and M.D.; writing—review and editing, F.T., R.A., and R.R. (Roberto Rondanelli); funding acquisition, M.D. and R.R. (Roberto Rondanelli). All authors have read and agreed to the published version of the manuscript.

**Funding:** This research was partially funded by the Fondecyt grants 1151476, 1151125, and 1161356. It was also partially supported by Conicyt Grant AFB180004 and BASAL Centre CATA PFB-06/2007. F.F. In addition, this material is based upon work supported by the Air Force Office of Scientific Research under award number FA9550-18-1-0249.

**Acknowledgments:** We would like to thank the strong support of Cristobal Garrido from Space and Planetary Exploration Laboratory and José Miguel Campillo from Department of Geophysics at University of Chile.

**Conflicts of Interest:** The authors declare no conflict of interest. The funders had no role in the design of the study; in the recollection, analyses, or interpretation of data; in the writing of the manuscript, or in the decision to publish the results.

## Abbreviations

The following abbreviations are used in this manuscript:

DAS     Data Acquisition System
FFT     Fast Fourier Transform
FMCW    Frequency-Modulated Continuous-Wave
IF      Intermediate Frequency
RF      Radio Frequency
LO      Local Oscillator
UAV     Unmanned Aerial Vehicle
VCO     Voltage Controlled Oscillator

## Appendix A. Implementation Specific Details of the Main Radar Blocks

### Appendix A.1. Transmitter Chain Details

The Gunn oscillator generates the radar waveform, which is transmitted to the environment through the transmitter antenna. The oscillator was recycled from the Cosmic Background Imager (CBI) radio-telescope [30]. The frequency of the oscillator is controlled with electronic resonators composed by a tunable microwave cavity and a YIG sphere. The cavity is set to select the frequency of the carrier wave, and the YIG sphere is used to modulate the output frequency with a voltage input. Adjusting the cavity, 38.6 GHz was the closest stable frequency to the desired 35 GHz.

A direct digital frequency synthesizer modulates the Gunn oscillator output frequency using a 150 kHz rising sawtooth voltage signal, yielding a non-ambiguous range of 500 m (see Equation (1), with $F_d = B$). The amplitude of the sawtooth is set to obtain a modulation bandwidth of 10 MHz, and thus producing, at best, a range resolution of 15 m (see Equation (2)).

The directional coupler after the Gunn oscillator allows the use of the transmitted waveform signal also as a local oscillator (LO) for the receiver mixer. The microwave insulator after the coupler avoids reflections entering back the transmitter antenna.

### Appendix A.2. Antenna Details

Both transmitter and receiver antennas are identical corrugated conical horns, also recycled from the CBI radio telescope [30]. The horns' antennae have an aperture of 8.1 cm and a 15° semi flare angle. Because these types of antennas have very little losses, their gain can be approximated to their directivity. The directivity for 35 GHz is estimated in 22.7 dB via simulations performed with ANSYS HFSS. The far-field region distance is estimated in 1.53 m, according to the wavelength and the antenna aperture size.

### Appendix A.3. Receiver Chain Details

The Receiver first component after the receiver antenna is a 20 dB low-noise amplifier (LNA). The amplified signal is multiplied with the transmitted waveform by a double balanced frequency-mixer. The output of the mixer (IF—10 MHz) goes through a chain of amplifiers with a gain of 64 dB, being terminated with a matched load of 50 Ω.

The chain of amplifiers after the IF output brings the signal to a level that better uses the dynamic range of the analog-to-digital-converter (ADC) of the Data Acquisition System (DAS). The maximum output power of the last IF amplifier avoids overcoming the maximum input voltage of the DAS.

The matched load termination allows sampling the IF signal with a high input impedance DAS. The analog bandwidth of the DAS (17 MHz) acts as a low-pass filter, reducing the noise power of the receiver.

*Appendix A.4. Data Acquisition System Details*

The data acquisition system (DAS) is based on PCI-DAS4020/12 by MCCDAQ, on-board of a desktop computer, and controlled by a LABVIEW script.

The DAS samples the IF voltage signal ($S_0$ in Figure 3) and the oscillator reference voltage signal ($S_1$ in Figure 3) simultaneously, using two independent 12-bit precision analog-to-digital-converters (ADC), acquiring $10^7$ samples, every 10 s for cloud measurements or 3 s during the drone experiment, with a sample-rate of 20 MHz, and with a range of $\pm 1$ V.

The digital output is transferred to random-access-memory (RAM) and then stored into a binary file inside a Hard Disk Drive (HDD). The RAM capacity limits the maximum amount of samples captured at once and the HDD writing speed limits the continuous acquisition of data at higher sampling rates.

It was noted that, when the acquisition system is used at the maximum sampling rate (20 MHz), noise spikes can appear randomly. These noise spikes can cause a loss of approximately 10% of the estimated profiles.

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
