# Peer review of "Low-Cost Ka-Band Cloud Radar System for Distributed Measurements within the Atmospheric Boundary Layer"

_remotesensing, doi:10.3390/rs12233965_

Round 1

Reviewer 1 Report

It is recommended to include the following comments.

major comments

The author showed the results of comparative cloud-base heights estimated from the developed radar and those observed from the ceilometer(Figure 8). One of the best ways to claim that the radar is properly developed(&calibrated) and ready for the operation would be to provide the results from the field experiment. However, there seems to be insufficient data showing the results observed in the actual field. It is recommended that the authors add things that could support the developed radar actually detects clouds well(eg., In addition to the cloud base, it seems to be necessary to add information on how well the cloud structure is observed).

Further, in the manuscript, the author showed the information of cloud-base, not the cloud structure (Figure 8). What would be the merits of using a developed cloud radar instead of using a ceilometer? Also, please explain how the cloud-base information can contribute to the purpose/goal of the study.

minor points

- page 10 of 14 range cell -> range gate

- Please change the location of Table 1 and Figure 6 (e.g., Figure 6 should be shown prior to Table 1)

- Line 142: polluted -> contaminated

- Lines 171-174: It seems that the FMCW radar is also not able to detect the cloud top.

- Line 198-199: What is the main purpose of the developed cloud radar? Is the purpose to detect the cloud-base heights? If then, what is the strength (merits) of radar over the lidar?

Author Response

Before going over the detailed concerns, we wnat to thank the revier for the concrens, questions and suggetsions, they really helped to improve the article. 

Concern #1: The author showed the results of comparative cloud-base heights estimated from the developed radar and those observed from the ceilometer (Figure 8). One of the best ways to claim that the radar is properly developed(&calibrated) and ready for the operation would be to provide the results from the field experiment. However, there seems to be insufficient data showing the results observed in the actual field. It is recommended that the authors add things that could support the developed radar actually detects clouds well (eg., In addition to the cloud base, it seems to be necessary to add information on how well the cloud structure is observed).

Response to concern #1: Thank you for the comment, it is a relevant point. For reasons escaping our control, since October last year it has been impossible to move for the country to repeat or extend our field campaign. In addition, Chile has no cloud radar, therefore without an instrument to compare or to confirm what we see with our radar inside a cloud, we decided to leave internal information out of the document and focus on showing that our instrument could replicate the measurements of another instrument that we actually have access to, a ceilometer.  However, in this new version we have included uncalibrated Doppler measurements (sections 3.1 and 4) as well as averaged profiles (reflectivity and Doppler). In addition, we have included data from other instruments near the area of the field campaign. In the discussion section (section 5) we have added data from a sounding launched by the  Dirección Metereológica de Chile at 12 UTC from Santo Domingo station about 120 km south of the field campaign and 2 hours earlier than our measurements. We also include MODIS data that confirms that the cloud structure is pretty similar during this region and period of time. With this data we can tell it is very likely that  the thickness of the cloud is approximately 300-350 m. Thrusting the lower boundary provided by the ceilometer the top might be at 700-750 m. This confirms that the ceilometer is not seeing the top of the cloud. Although we have some echoes from 600km with our radar, these echoes are over the estimated range limit, therefore the echos are weak, this might be due to the fact that the reflectivity inside clouds usually increases with height.  Since this campaign was designed to test the range limit of the radar comparing it with a ceilometer instrument, we consider this a strong demonstration that the radar is operative. However, extra work (improving calibration) and campaigns are necessary to show that this instrument can be a good complement to ceilometer measurements as next steps. Although, part of it should be performed outside Chile since we do not have radar capabilities/facilities in Chile. We hope to be able to advance in that direction as soon as possible.       

Concern #2: Further, in the manuscript, the author showed the information of cloud-base, not the cloud structure (Figure 8). What would be the merits of using a developed cloud radar instead of using a ceilometer? Also, please explain how the cloud-base information can contribute to the purpose/goal of the study.

Response to concern #2: Thank you for the question, with it we realized that the main focus of our preliminary demonstration was not clear. We think that we have improved our justification of the work done thus far in the introduction and in the discussion sections. Although the radar has some advantages of cost, portability and power consumption compared to the ceilometer, which might facilitate the multi point study of the base evolution with time in the coastal range region of Chile. Nevertheless, the ceilometer provides more information on the state of development of our radar. But, the main focus of this work was to show that the system is acceptably working to detect the base of the cloud similarly to the ceilometer, which is the instrument we had access to compare our instrument with. With these we expected to confirm the range limit and resolution of our radar. Confirming this, we can have some confidence that the radar might be properly working as a cloud detector, thus justifying the next step of development, which includes the calibration of Doppler and cross comparison with other radars outside Chile. In our logic, if we are not able to replicate the operation of other more robust instruments it is unlikely that we might be doing well in new territory, such as measuring internal structure of clouds. Thus, this work is to confirm that we can be able to detect clouds up to 500 m of altitude with a resolution of 30m or better, which justify the next steps of development. We hope this explanation and the changes in the paper reflects this.      

Minor comments:

- page 10 of 14 range cell -> range gate 

Thanks for noticing. It is a good suggestion and we performed the change in the new version of the article. We consolidated the term to range cell in the whole document as suggested. 

- Please change the location of Table 1 and Figure 6 (e.g., Figure 6 should be shown prior to Table 1) 

Thanks for noticing the issue. The change was already implemented. In the latex Fig 6 is called before Table 1, however, latex optimization is changing the order to avoid white spaces in the documents. We use [H] argument to force the order but it might change after editing of the journal.

- Line 142: polluted -> contaminated 

Thanks for the suggestion. It was accepted and changed in the document. 

- Lines 171-174: It seems that the FMCW radar is also not able to detect the cloud top. 

Thanks for noticing this issue. We understand now that the inclusion of this paragraph in the discussion section produced a confusion that our radar was able to detect the top of the cloud. This paragraph was intended to justify why a radar might be a complement to a ceilometer. We express in a clearer manner this time (with the inclusion of more data from MODIS and radiosonde near the area) that our radar was not able to detect the top of the cloud. To avoid confusion, we moved this paragraph to the introduction as an explanation of why this type of radars (including short range ones) can complement the work of ceilometers.    

- Line 198-199: What is the main purpose of the developed cloud radar? Is the purpose to detect the cloud-base heights? If then, what is the strength (merits) of radar over the lidar? 

Thanks for this comment. We found this comment very similar to concern #2 (see above). We apologize if we misunderstood the comment, be we think that the response to concern #2 is also appropriate to this comment. However, this is so relevant that we will comment again shortly here. The main focus of the work is to confirm that the radar works similar to a ceilometer to detect the base of the radar. We also wanted to verify the range limit and resolution of the radar, by comparing the data of the radar with data obtained with a ceilometer. This can give us confidence in the radar operation to continue its development, calibration and evaluation to provide internal information of clouds. Unfortunately, the work has been stopped due to many internal (Chile) and external events, due to limitations in movement within the country and to movie equipment outside Chile.

Reviewer 2 Report

This paper details improvements to a low-cost, Ka-band FMCW radar for measurements of coastal fog and low clouds. There is certainly merit to innovation of low SWaP and low cost cloud and precipitation radars for deployment in a distributed ground-based network, or for mounting on UAVs for autonomous operation. The system design and radar front- and back-ends are sufficiently discussed, and there are no technical errors in the description of the hardware or the FMCW measurement principle. However, there are several significant issues with the data analysis and observations presented in this work, and some of the conclusions are not supported by the measurements. Some of the issues with the manuscript are as follows:

  1. (minor) The authors use the term "spatial resolution" to mean range resolution. This is confusing to the reader since it is unclear "spatial resolution" refers to the horizontal resolution defined by the beam footprint, or the range resolution defined by the chirp bandwidth.
  2. (major) Stratus and fog often result in very low radar echo power values with low signal-to-noise ratios (SNR). The authors do not discuss whether or not they subtract off the contribution to the detected power of the system noise power. If this is not done, the estimated reflectivities become significantly biased (high) in the low-SNR regime.
  3. (major) The authors implement an unconventional signal processing technique in terms of FMCW radars. Specifically, the lag-1 autocorrelation used here assumes that sequential pulses are highly correlated. This is a common technique in pulsed radars for estimated the mean Doppler velocity, but the authors do not justify this choice of signal power estimator by proving that sequential pulses are correlated. Typically, in FMCW radar, one obtains the echo power spectrum by taking the modulus squared of the variable "R" (i.e. lag-0 autocorrelation) and performing an incoherent average of this quantity for N pulses at each range. Have the authors taken care to ensure that the pulse repitition interval is much less than the decorrelation time? This must be shown.
  4. (major) The authors claim that an advantage of a cloud radar over ceilometer is that the radar can detect the cloud top while the lidar cannot. However, this conclusion is not supported by the measurements presented. In figure 8, the apparent thickness of the low cloud layer is much larger in the ceilometer plot than in the radar plot.

Author Response

Before going over the detailed concerns, we wnat to thank the revier for the concrens, questions and suggetsions, they really helped to improve the article. 

General comment: The main strategy of this work is to develop a simpler Basta FMCW radar. Therefore, we follow its development. We are pretty similar except for the fact that we conceived our system to be portable and lower in cost. In addition our analog front end is simpler putting the weight on the digital processing (FPGA). The first goal that we need to be accomplished is to prove that we are detecting clouds (at least qualitatively speaking) with the expected range limit and resolution.

Concern #1: (minor) The authors use the term "spatial resolution" to mean range resolution. This is confusing to the reader since it is unclear "spatial resolution" refers to the horizontal resolution defined by the beam footprint, or the range resolution defined by the chirp bandwidth.

Response concern #1: Thanks for bringing attention over this terminology issue. The term “spatial resolution” was changed for “range resolution” from the article (abstract and introduction), since we are only referring to range resolution in this work.  

Concern #2: (major) Stratus and fog often result in very low radar echo power values with low signal-to-noise ratios (SNR). The authors do not discuss whether or not they subtract off the contribution to the detected power of the system noise power. If this is not done, the estimated reflectivities become significantly biased (high) in the low-SNR regime.

Response concern #2: Thank you for the comment, it let us know that our description was not clear or explicit enough about this topic. We do a process similar to that performed by Basta radar (reference [12] in the article), in this radar the noise is assumed coming from the last range gates plus 2 dB, however it is not subtracted, it is only used as a mask. Determination of the noise to improve the quantitative precision of our radar is the next step. In this work we wanted to expose to the community that the radar is detecting clouds. It justifies the refinement of the procedures, in particular calibration to proper estimation of noise level.     

Concern #3: (major) The authors implement an unconventional signal processing technique in terms of FMCW radars. Specifically, the lag-1 autocorrelation used here assumes that sequential pulses are highly correlated. This is a common technique in pulsed radars for estimated the mean Doppler velocity, but the authors do not justify this choice of signal power estimator by proving that sequential pulses are correlated. Typically, in FMCW radar, one obtains the echo power spectrum by taking the modulus squared of the variable "R" (i.e. lag-0 autocorrelation) and performing an incoherent average of this quantity for N pulses at each range. Have the authors taken care to ensure that the pulse repetition interval is much less than the decorrelation time? This must be shown.

Response concern #3: Thank you very much for the question. This is a relevant topic and probably we could explain it better in the previous version of the article. We use lag-1 approach, since our SNR is low and this procedure reduces the noise to a level where we see the target. It is the same approach that Basta radar uses. Regarding the concern of the oscillator stability, we performed an experimental evaluation of the oscillator stability, where for a fixed control voltage, after allowing the oscillator to reach a steady temperature we measured the frequency variation (using a spectrum analyser), which reached 8 kHz in a period of 80μs. This is acceptably slow given the repetition period of 6.6μs. This information is now explicitly included in the first paragraph of section 2.3 (Digital processing). 

Concern #4: (major) The authors claim that an advantage of a cloud radar over ceilometer is that the radar can detect the cloud top while the lidar cannot. However, this conclusion is not supported by the measurements presented. In figure 8, the apparent thickness of the low cloud layer is much larger in the ceilometer plot than in the radar plot. 

Response concern #4: Thanks for highlighting this issue. We understand now that the inclusion of this paragraph in the discussion section transmitted the wrong message that we were referring to our radar, implying that our radar was able to detect the top of the cloud. This paragraph was intended to justify why a radar might be a complement to a ceilometer. We express this in a clearer manner this time (with the inclusion of more data from MODIS and radiosonde near the area) highlighting that our radar was not able to detect the top of the cloud. To avoid confusion, we moved this paragraph to the introduction as an explanation of why radars (including short range ones) can complement the work of ceilometers.

Reviewer 3 Report

This paper describes a compact, low power and low-cost Ka-band FMCW radar for measurements of fog and low level marine stratus for Chilean costal range. The radar hardware was built based on recycled radio astronomy components in order to keep the cost low.  A Gunn oscillator with 13 dBm output power at 38 GHz was used as its transmitter. Range calibration using a UAV drone was carried out. Radar performance was verified by comparing measurements of a thin layer of low-level clouds to data from a collocated lidar ceilometer.

The authors are reporting on their low-cost system that may have cost effective applications in Chile. Therefore, it is worthy of publication with improvements in the technical discussion. The radar hardware itself is not new for research.  The technical discussion in the paper lacks some details such as receiver noise figure, digital sampling rate, integration time, etc.  Also, some of the discussion on evaluation of the radar performance lacks details such as range resolution and sensitivity.  Also, some of the range calibration steps are not appropriate, such as range resolution is determined by counting range gate number between two calibration target positions..

 Comments,

  1. The discussion on range calibration using a UAV is not well described.
  • The paper mentioned that the FMCW radar chirp bandwidth is 10 MHz. According to Eq (2), range solution should be 15 m (Appendix A.1, Lines 267-268), but the authors also drew conclusion that radar range solution is 20 m based on calibration measurements from a UAV target, different from theoretically estimated resolution of 30 m (lines 134 and 201). The authors seemed to estimate radar range resolution by counting gate numbers between two known target positions. This approach is not appropriate.  
  • What is the digital receiver sampling clock? This will determine the radar signal sampling gate spacing, but not the true range resolution.
  • Table 1 gives that the height of the calibration target – a UAV drone was at 200 m, 300 m, and 400 m during the calibration. How were these height values determined relative to the radar antennas?
  • Table 1 and Figure 1 show that the measured uncorrected UAV ranges are 381.4 m, 528.1 m, and 674.8 m, So the height differences among these three positions are 528.1-381.4 = 146.7 m, and 674.8-528.1 = 146.7 m, not the same as the nominal height difference of 100 m, why? Is the true chirp bandwidth 10 MHz?
  • How the 60 m range offset was estimated (Lines 136-137)?
  • In Table 1, the measured corrected UAV positions are 100+/- 2 m, 200 +/-3 m, 300+/- 3 m. How the +/- 2 m and +/- 3m uncertainties are determined?

  1. The paper did not provide time-range profile to show the receiver noise floor. Instead, radar sensitivity was given based on a theoretical estimate and limited measurement data. What is the receiver noise figure? How was the -55 dBZ sensitivity estimated? What is the integration time? With a 150 KHz PRF, the radar digital receiver should have accumulated data from many time-range profiles. Was integration over time performed in order to improve the SNR? (Figure 8, With 30 minutes observation (11:00-11:40) and a 150 KHz PRF, the radar should have recorded 30*60*150000 = 2.7 x10^6 profiles)

  1. Figure 8, radar image shows pixels up to 600 m while the ceilometer with much finer resolution only show a cloud layer between 400 m and 500 m. Are these radar pixels between 500 m and 600 m real cloud targets?

  1. Confusion in symbol usage.
  • In Eq (2), DR stands for range resolution; In Eq (7) and Table 2, Rres is used for range resolution.
  • In Eq (1), r stands for range; In Line 83, Rx stands for radar receiver; Eq (4) & (5), In Lines 101 & 104, R represents impulse response.
  • In Eq (1), c stands for speed of light; In Eq (3), C stands for calibration constant; Between Lines 83 and 104, Figure 3, C0 and C1 stand for IF and chirp modulation signal, respectively; In Figure 3, C = -9.1 dB stands for coupling factor.

Author Response

Before going over the detailed concerns, we wnat to thank the revier for the concrens, questions and suggetsions, they really helped to improve the article. 

Concern #1: The discussion on range calibration using a UAV is not well described. The paper mentioned that the FMCW radar chirp bandwidth is 10 MHz. According to Eq (2), range solution should be 15 m (Appendix A.1, Lines 267-268), but the authors also drew conclusion that radar range solution is 20 m based on calibration measurements from a UAV target, different from theoretically estimated resolution of 30 m (lines 134 and 201).

Response concern #1: Thanks for the comment. We have removed the allusions to the 30 m range resolution to avoid confusion. The 15 m is the theoretical value obtained by using the whole ramp. We do not use the whole ramp to avoid aliasing. The theoretical range resolution using the half of the ramp is 30m. The 20 m resolution is obtained by calibrating by the drone. The 20 m is the fiablest value, since it was estimated from actual measurements. We have changed the explanation in section 3.1 (Distance Calibration with an Unmanned Aerial Vehicle) last paragraph to clarify this point. 

Concern #2: The authors seemed to estimate radar range resolution by counting gate numbers between two known target positions. This approach is not appropriate.

Response concern #2: Thanks for requesting clarification of this topic. We have improved the procedure and we have also cited the reference in which we are based (reference [17]). In this procedure we do a linear fit with the 3 points that relates range with frequency, as described in [17].  

Concern #3 What is the digital receiver sampling clock? This will determine the radar signal sampling gate spacing, but not the true range resolution.

Response concern #3: Thank you for the question. The sampling clock of the receiver’s digital stage is 20 MHz, which in fact defines the sampling  gate spacing. On the other hand, indeed the range resolution depends on the bandwidth of the radar. We think this is described in the previous reference cited in the article therefore we did no action this time in the article. Although it is possible we misunderstood the concern. 

Concern #4: Table 1 gives that the height of the calibration target – a UAV drone was at 200 m, 300 m, and 400 m during the calibration. How were these height values determined relative to the radar antennas?

Response concern #4: Thanks for noticing this lack of information. The UAV has a GPS+GLONASS GNSS geolocation receiver. It was used to estimate the altitude. The system estimates the altitude subtracting the altitude at the take off level (on rest). We have included this information in 3.1 (Distance Calibration with an Unmanned Aerial Vehicle), Figure 5.  

Concern #5: Table 1 and Figure 1 show that the measured uncorrected UAV ranges are 381.4 m, 528.1 m, and 674.8 m, So the height differences among these three positions are 528.1-381.4 = 146.7 m, and 674.8-528.1 = 146.7 m, not the same as the nominal height difference of 100 m, why? Is the true chirp bandwidth 10 MHz?

Response concern #5: Thanks for the question. Our justification for this is that the VCO control voltage slightly changed (which changes the bandwidth of the chirp) since the configuration to the time when the experiment was performed. This is one of the main reasons to perform the calibration, since it is hard to set up all the elements in the system with high precision. Calibration helps in removing these types of systematics errors. 

Concern #6: How the 60 m range offset was estimated (Lines 136-137)? In Table 1, the measured corrected UAV positions are 100+/-2 m, 200 +/-3 m, 300+/- 3 m. How the +/- 2 m and +/- 3m uncertainties are determined?

Response concern #6: Thank you for noticing this typo. The third line should not be visible. It was left there by error from some preliminary version to display the data. Table 1 was changed for clarity.  

Concern #7: The paper did not provide time-range profile to show the receiver noise floor. Instead, radar sensitivity was given based on a theoretical estimate and limited measurement data. What is the receiver noise figure? How was the -55 dBZ sensitivity estimated?

Response concern #7: Thank you for highlighting this topic. We haven’t  performed experimental measurements of the noise. The noise was estimated from the penultimate gate where no noise is expected. We were able to perform a theoretical estimation of the temperature equivalent of the system noise (using microwave studio) which is estimated close to 1450 K. Although this does not include other possible sources of noise, such as the atmosphere. We use -55 dBz since from our laboratory correction (which is not very accurate) we were seeing a cloud with -55 dBZ at 400 m. Calibration Is key to improve these estimations. We have improved the discussion section to include these concerns. 

Concern #8: What is the integration time? With a 150 KHz PRF, the radar digital receiver should have accumulated data from many time-range profiles. Was integration over time performed in order to improve the SNR? (Figure 8, With 30 minutes observation (11:00- 11:40) and a 150 KHz PRF, the radar should have recorded 30*60*150000 = 2.7 x10^6 profiles)

Response concern #8: Thank you for the question. Integration was used to improve the SNR. The radar can continuously acquire data up to 1 second each 10 seconds. We coherently integrate only the measurements continuously taken. The integration details are in the annexe A.4. 

Concern #9: Figure 8, radar image shows pixels up to 600 m while the ceilometer with much finer resolution only show a cloud layer between 400 m and 500 m. Are these radar pixels between 500 m and 600 m real cloud targets?

Response concern #9: Thank you for the question, it is a really good question. We think these echoes are part of the cloud, although as the Lidar cannot reach that altitude we only have estimation from MODIS and sounding, which provide an estimate of the top of the cloud in the range  650 and 750 m.  It would be really good to place our radar next to other more trusted radar. We hope this to be the next step, together with a more precise calibration. We have improved the discussion section to extend this subject.    

Concern #10: In Eq (2), ΔR stands for range resolution; In Eq (7) and Table 2, Rres is used for range resolution.

Response concern #10: Thank you for noticing the excess of notation. We have changed the range resolution to Δr and we are consistent in its use in Eq. (2), Eq. (7) and Table 2. 

Concern #11: In Eq (1), r stands for range; In Line 83, Rx stands for radar receiver; Eq (4) & (5), In Lines 101 & 104, R represents impulse response.

Response concern #11: Thank you for noticing the excess of notation. We have removed the Rx reference along the article. Although we kept r and R as originally presented since we think they are different enough to avoid confusion. In addition, with this notation we are consistent with reference [12] (Basta radar) notation, which might facilitate the understanding of the readers. 

Concern #12: In Eq (1), c stands for speed of light; In Eq (3), C stands for calibration constant; Between Lines 83 and 104, Figure 3, C0 and C1 stand for IF and chirp modulation signal, respectively; In Figure 3, C = -9.1 dB stands for coupling factor. 

Response concern #12: Thank you for noticing the excess of notation. We have changed the calibration constant notation to Cradar to avoid confusion. C0 and C1 were changed to S0 and S1 respectively. Finally the coupling factor in Figure 3 is now represented by CF. Only c was kept as originally as the speed of light.

Round 2

Reviewer 2 Report

Thank you to the author's for responding to my comments. While it is clear the authors have edited the manuscript considerably since the previous submission, a few minor issues remain with this manuscript.

  1. It is incorrect to convert the raw detected power to reflectivity units (dBZ) without subtracting off the noise floor, especially in the low-SNR regime where the authors operate. If the authors only want to make the point that they can detect weak clouds/fog, they can simply display the detected signal in arbitrary dB (or calibrated dBm), but it is important to clarify that this quantity represents the radar echo power PLUS the system noise power. And thus is not proportional to the cloud reflectivity.
  2. The author's did not fully address my comment about pulse-to-pulse coherence, as I was specifically referring to decorrelation due to random motion of the hydrometeor targets (e.g. Doppler velocity spectrum width). However, now that I see that the radar PRI is ~ 6 us, it is clear that sequential pulses are going to be highly correlation (provided the oscillator is stable enough, which the authors have provided evidence for). It should be very clear in the main manuscript what were the radar parameters used for signal acquisition. These include at minimum the pulse time and chirp bandwidth. By looking closely I can see that the PRF is included as part of the block diagram, but this should be specifically called out in the text or a table, and the expected range resolution should be described in the main text (not just in an appendix).
  3. On a related point, it is a bit concerning that the authors describe a chirp bandwidth of 10 MHz and associated range resolution of 15 m in the appendix, yet their results indicate a range resolution of 20 m. This is a large discrepancy, and reasons for this inconsistency are not given.

Author Response

We are uploading (a) our point-by-point response to the comments (below) (response to reviewer #2), and (b) an updated manuscript with the changes derived from the reviewers’ comments and suggestions. We really want to thank the reviewer for the effort to review our work again and we think her/his comments and suggestions continued improving the new version of the article.

In this document, we present each review. We comment on each reviewer’s concern. The reviewer comments are in regular text, while the responses are in italic text.

Comments Reviewer #2

Thank you to the author's for responding to my comments. While it is clear the authors have edited the manuscript considerably since the previous submission, a few minor issues remain with this manuscript.

Concern #1: It is incorrect to convert the raw detected power to reflectivity units (dBZ) without subtracting off the noise floor, especially in the low-SNR regime where the authors operate. If the authors only want to make the point that they can detect weak clouds/fog, they can simply display the detected signal in arbitrary dB (or calibrated dBm), but it is important to clarify that this quantity represents the radar echo power PLUS the system noise power. And thus is not proportional to the cloud reflectivity.

Response concern #1: Thank you very much for your concern. We agree that the noise removal is a relevant procedure. Since we do not have a precise estimation of the noise, we used the mask power  (threshold) as the noise level of the system, which was removed from each power value over the mask threshold. This signal processing step is now described in the last paragraph of section 2.3 (Digital Processing - line 146). This procedure reduced the reflectivity estimation by approximately 5 dBZ (at the cloud base). We updated all references regarding measured reflectivity values during the field campaign. We also updated figure 8(a) to present reflectivity obtained after subtracting the noise power to the raw detected power. Finally we made some changes in the Discussion section due to the changes in Figure 8(a) (paragraph starting in line 236).

Concern #2: The author's did not fully address my comment about pulse-to-pulse coherence, as I was specifically referring to decorrelation due to random motion of the hydrometeor targets (e.g. Doppler velocity spectrum width). However, now that I see that the radar PRI is ~ 6 us, it is clear that sequential pulses are going to be highly correlation (provided the oscillator is stable enough, which the authors have provided evidence for). It should be very clear in the main manuscript what were the radar parameters used for signal acquisition. These include at minimum the pulse time and chirp bandwidth. By looking closely I can see that the PRF is included as part of the block diagram, but this should be specifically called out in the text or a table, and the expected range resolution should be described in the main text (not just in an appendix).

Response concern #2: Thank you very much for your comment. We appreciate that although we were not that clear, as you mentioned in your comment, to the previous concern we were able to reduce the doubts about the correlation of the pulses, therefore, the  property of the used methodology.  On the other hand, we agree that those parameters are important and they should be easily found in the article. We added details of the radar parameters used for signal acquisition and the expected range resolution in the main text, specifically in the second and third paragraphs of section 2.2 (Radar Hardware) - lines 108 and 113 respectively.

Concern #3: On a related point, it is a bit concerning that the authors describe a chirp bandwidth of 10 MHz and associated range resolution of 15 m in the appendix, yet their results indicate a range resolution of 20 m. This is a large discrepancy, and reasons for this inconsistency are not given.

Response concern #3: Thank you for highlighting this issue. The radar resolution depends primarily on the chirp bandwidth. However, the actual range resolution can be degraded due to non-ideal factors, such as reduced effective modulation bandwidth due to time-of-flight delays, the increased main-lobe null-to-null bandwidth when FFT windows are applied, a reduced receiver frequency resolution due to a reduction in the pulse processing time, and non-linearity of frequency sweeps [15] [16]. Thus, the theoretical range resolution is the best achievable resolution. This issue highlights the need of the external calibration method for the range (resolution and maximum achievable). We added the provided arguments and cites to the article in lines 88 and 89 of Section 2.1 (Measurement Principle) and second paragraph of the Section 5 (Discussion), starting in line 210.

This manuscript is a resubmission of an earlier submission. The following is a list of the peer review reports and author responses from that submission.

Round 1

Reviewer 1 Report

This paper covers the use a cloud radar and its performance was benchmarked with the air of a lidar and an instrumented drone. The paper is appealing and needs some polishing in the language with some grammar errors to be corrected. On the scientific side the paper is well introduced and described but I found the performance tests could be well improved on the discussion side and this section is very short and should be improved. I understand sometimes some budgetary issues appear on research but the authors should be creative to bypass the calibration problems and point when they should be taken into consideration. 

Reviewer 2 Report

This paper doesn't include any novel research, and none of the results presented appear to be publishable. The authors have padded the length of the paper by describing, at length, very basic radar principles which could be gleaned from any undergraduate textbook on radar fundamentals. The FMCW radar described herein fails to accurately or consistently predict range estimates for both a calibration target and a water vapor cloud. The reasons given for this inaccuracy is ambiguous, unconvincing, and only very briefly discussed. The false alarm rate of the system seems to be extremely high, but no effort has been made to characterize this false alarm rate.

I don't believe this work can be published until an entirely new experiment is conducted after the kinks are worked out of the current system, and until some novel measurement technique or algorithm is developed.

Reviewer 3 Report

The paper requires minor editing, including improvement of grammar and word choice.